# Influence of Serum Vitamin D Levels on Survival Rate and Marginal Bone Loss in Dental Implants: A Systematic Review

**DOI:** 10.3390/ijerph191610120

**Published:** 2022-08-16

**Authors:** Santiago Bazal-Bonelli, Luis Sánchez-Labrador, Jorge Cortés-Bretón Brinkmann, Carlos Cobo-Vázquez, Natalia Martínez-Rodríguez, Tomás Beca-Campoy, Juan Santos-Marino, Emilio Rodríguez-Fernández, Mario Alvarado-Lorenzo

**Affiliations:** 1Department of Dental Clinical Specialties, Faculty of Dentistry, Complutense University of Madrid, 28040 Madrid, Spain; 2Private Practice, 28006 Madrid, Spain; 3Department of Surgery, Faculty of Medicine, University of Salamanca, 37008 Salamanca, Spain; 4Postgraduate Program in Oral Surgery, Implant and Restorative Dentistry, University of Cordoba, 14004 Cordoba, Spain; 5Postgraduate Program in Oral Surgery, Periodontics and Restorative Dentistry, San Antonio de Murcia Catholic University, 30107 Murcia, Spain

**Keywords:** dental implant, vitamin D, osseointegration, bone resorption

## Abstract

This systematic literature review set out to investigate the relationship between serum vitamin D levels and dental implants in terms of survival rates, marginal bone loss, and associated complications. The review was conducted according to PRISMA guidelines, performing an electronic search in four databases (Pubmed, Web of Science, Cochrane, and Scopus), complemented by a manual search up to April 2022. Four articles were selected for analysis. The Newcastle-Ottawa Quality Assessment Scale tool was used to assess the quality of evidence of cohort studies, and the Cochrane bias assessment tool was used to assess the quality of evidence of randomized clinical trials. The study included 1089 patients restored with 1984 dental implants, with follow-up periods ranging from 20–240 months. Cases presenting lower serum vitamin D levels obtained slightly worse results in terms of marginal bone loss. Longer follow-up periods are needed in order to determine whether serum vitamin D levels affect implant survival rates and osseointegration over time.

## 1. Introduction

Dental implants are a successful means of restoring both dental function and esthetics in edentulous and partially edentulous patients [1,2]. Dental implant-based treatments to replace missing teeth enjoy predictable outcomes and high long-term success rates [3,4].

For osseointegrated implant-based treatments, the criteria that ensure success include the formation of direct bone to implant contact (BIC), primary stability, and the development of functional implant ankylosis or osseointegration [5,6]. Successful osseointegration is influenced by several factors [7] including surgical and prosthetic factors [3,8]; factors related to the implant itself such as material, design [9], and surface topography; the time elapsed since implant placement [10]; and patient-related factors such as the host bone’s quality, quantity, and response to implantation [3,11,12]. Most research has focused on surgical, prosthetic, and implant-related factors [9,13] and has generated an ever-growing body of knowledge, understanding, and experience. Nevertheless, implant osseointegration failures continue to occur [14], which suggests that perhaps there are additional patient-related risk factors that need to be considered and investigated [15].

Vitamin D is a fat-soluble molecule obtained from the diet or generated in the skin by exposure to ultraviolet light [16,17]. Vitamin D in its active form (1,25-dihydroxy vitamin D3) plays a key role in bone mineral homeostasis by stimulating intestinal absorption of calcium and phosphate [18]. Vitamin D deficiency can lead to bone injury, such as fracture and bone loss, and severe vitamin D deficiency dramatically increases the risk of infection and even mortality (Javed et al., 2016). Vitamin D levels are usually classified into three different groups: a serum vitamin D level <10 ng/mL is considered deficient, in the range of 10–30 ng/mL insufficient, and >30 ng/mL sufficient [19].

Since vitamin D is involved in bone metabolic processes and regulates the immune system; it is currently of special interest to dentists placing dental implants. There is some evidence that the correct concentration of this prohormone correlates positively to the dental implant osseointegration process. Numerous studies have shown that vitamin D plays a potentially important role in the postsurgical tissue repair process, in the integration of the implant into bone tissue, and homeostasis around the implant after loading with a prosthetic restoration [19,20].

To date, no systematic review has investigated the relationship between serum vitamin D levels and dental implant outcomes, including only human studies. Therefore, the present review aimed to address the following questions: is there a relationship between serum vitamin D levels and implant survival rates? Does marginal bone loss increase with low vitamin D levels? Are there further intraoperative and postoperative complications that depend on serum vitamin D levels?

## 2. Materials and Methods

### 2.1. Review Development and Focused Question

The present review was registered in the PROSPERO database (Nº: CRD4202232320121). It was conducted according to the guidelines of the PRISMA (Preferred Reporting Items for Systematic Review and Meta-Analyses) statement [21] and the PICO (Population, Intervention, Comparison, Outcome) scale was used for the design:

*Population.* Systemically healthy edentulous and partially edentulous patients;

*Intervention*. Implant placement with normal serum vitamin D levels;

*Comparison*. Implant placement with low serum vitamin D levels;

*Outcome.* Survival rate and marginal bone loss of placed implants.

Thus, the PICO question was: In edentulous and partially edentulous patients (P), what is the survival rate and amount of marginal bone loss (O) after placement of osseointegrated implants in patients with normal serum vitamin D levels (I) compared with patients with low serum vitamin D levels (C)?

### 2.2. Eligibility Criteria

#### 2.2.1. Inclusion Criteria

Human clinical studies that involved osseointegrated implants placed in patients with different serum vitamin D levels, randomized clinical trials, observational longitudinal prospective studies, observational longitudinal retrospective studies, case-control studies, and case series were included. Clinical human studies provided the following data: serum vitamin D levels, implant survival rate, and amount of marginal bone loss after follow-up of at least three months. The number of patients/study arm or cohort was >5 patients. Articles were published in English, Spanish or German. No restrictions were imposed on publication date.

#### 2.2.2. Exclusion Criteria

Clinical studies in which osseointegrated implants are placed, but serum vitamin D levels are not measured. Cross-sectional studies, animal studies, and case reports. In vitro studies.

### 2.3. Type of Intervention and Comparisons

The review analyzed osseointegrated implant placement procedures in patients with different serum vitamin D levels. Studies comparing low and normal serum vitamin D levels and studies investigating low serum vitamin D levels alone were included.

### 2.4. Data Collection

The primary outcomes analyzed to evaluate implant osseointegration were implant survival and marginal bone loss. Secondary outcomes were intraoperative and postoperative implant complications.

Dental implant survival was understood as the absence of mobility, without progressive marginal bone loss or infection leading to implant removal.

### 2.5. Sources and Search Strategy

The search strategy was to locate studies published in English, German, and Spanish dated before 14 April 2022, using the following search terms: “vitamin D”, “dental implant”, “survival rate”, “osseointegration”, and “marginal bone loss”. The databases consulted were: The National Library of Medicine (MEDLINE/Pubmed); Web of Science; Cochrane Library; and SCOPUS.

### 2.6. Study Selection and Screening Methods

Titles and abstracts were reviewed independently in the databases, with manuscripts meeting the inclusion criteria detailed above also being read by two reviewers (S.B.B and L.S.L), as well as papers without sufficient data in the title and abstract on which to base their decision.

Any disagreement between the two reviewers was resolved by a third reviewer (J.C.B.B). Projects with the longest follow-up period were selected. Inter-reviewer reliability of the full text was calculated (percentage of agreement and kappa correlation coefficient).

### 2.7. Clinical Data Extraction

Clinical data such as authors, year of publication, number of patients, mean age, number of implants, duration of follow-up, implant survival, and marginal bone loss were extracted from the manuscripts. In the case of incomplete data, the authors were contacted and data were omitted only when they were not available. It was performed by two independent reviewers and in duplicate (S.B.B and L.S.L).

### 2.8. Risk of Bias Analysis

The Cochrane Handbook of Systematic Reviews and Interventions version 5.1.0, which was applied to randomized clinical trials, was used to assess the risk of bias. The manual is divided into seven domains such as selection and allocation bias, blinding of participants and staff, outcome assessors, incomplete data, selective outcome reporting, and other biases. Each was classified as low risk, high risk or uncertain risk (represented by green (+), red (-) or yellow (?), respectively [22].

To assess the quality of other studies such as observational, cohort, non-randomized clinical trials, the Newcastle-Ottawa scale (NOS) was used, with a modified version for cross-sectional studies. This scale assesses study group selection, between-group comparison, and outcomes (maximum score of nine) [23].

## 3. Results

### 3.1. Study Selection

The initial database search located 81 articles in the MEDLINE/PubMed database, 76 titles in the SCOPUS database, 127 titles in the Web of Science database, and nine titles in the Cochrane Library. The manual search located another two articles. Of the total 452 articles, 180 were duplicates and were discarded. After an initial screening to identify articles irrelevant to the PICO question, followed by title and abstract screening, the full texts of 10 articles were read. Finally, four studies were selected for inclusion in the review and underwent data extraction and analysis. The flowchart shown in Figure 1 illustrates each step in the search and screening process. The articles excluded and the reasons for their exclusion are shown in Table 1 [24,25,26,27,28]. In this review, the study by Mangano et al. [29] was considered a continuation of the study by Mangano et al. [30], so it was treated as a single study for purposes of data extraction.

### 3.2. Study Characteristics

The four articles selected consisted of one retrospective study of three cohorts [29], two randomized clinical trials [31,32], and one prospective study of three cohorts [33]. All were published between 2017 and 2021. Table 2 provides information about the studies reviewed: study type, number of patients, patients’ gender, mean age, vitamin D serum levels, number and location of implants, implant survival, marginal bone loss, bone remodeling during osseointegration, and follow-up duration.

### 3.3. Quality Assessment

The NOS scale was used to assess the quality of the retrospective study of three cohorts [29] and the prospective study of three cohorts [33] The retrospective study scored seven points (low bias) [29] and the prospective study six points (medium bias) [33], with the most important biases being observed in the null demonstration that the outcome of interest was not present at baseline (S4) and in the null description of outcome assessment (E1) [29,33]. In addition, one of the studies did not have a sufficiently long follow-up for strong results to be observed (E2) [33]. These scores point to an adequate quality of evidence among these studies (Table 3). The RCTs were assessed with the Cochrane bias assessment tool scale. The risk of bias of one of the RCTs was medium [32], while for the other it was high [31]. High risk of bias was due to the allocation, concealment, and randomization domain, which was not carried out at all in Garg et al. [31]. At the same time, neither study [31,32] specified whether or not operators and participants were blinded, creating the risk of bias medium (Table 4).

### 3.4. Synthesis of Results

Inter-reviewer agreement

The Kappa statistic for inter-reviewer agreement (S.B.B and L.S.L) was 98%; kappa = 0.985; *p* < 0.001; 95% CI: 0.99; 0.95 for article screening, and 97%; kappa = 0.978, *p* < 0.001; 95% CI: 0.99; 0.52 for article analysis. Consultation with a third reviewer to reach consensus was not required.

Patient characteristics

The studies included a total of 1129 patients, whose age ranged from 20 to 57.3 ± 14.4 years [29,31,32,33] and comprised 529 women and 568 men, although in one article the sex of the participants was not stated [31].

Fifty-seven patients presented serum vitamin D levels <10 ng/mL, 478 patients levels between 10–30 ng/mL, 471 patients levels >30 ng/mL, 59 patients levels <30 ng/mL, and 64 patients levels <30 ng/mL, but supplemented by vitamin D.

A total of 1984 dental implants were placed in the 1129 patients with follow-up periods ranging from 3 months to 20 years [29,31,32,33].

Study results

Due to the low number of studies and the fact that not all the studies provided information on bone remodeling during osseointegration, implant survival, and marginal bone loss, meta-analysis could not be carried out; instead, a qualitative synthesis of the results was performed.

In the three-cohort restrospective study by Mangano et al. [29] including a total of 885 patients and 1740 dental implants, the three cohorts were divided into groups according to serum vitamin D level and evaluated two weeks before implant placement into deficient (<10 ng/mL), insufficient (10–30 ng/mL), and satisfactory (>30 ng/mL). At the 14-year follow-up, the authors found that dental implant failure in the satisfactory group was 2.9%, in the insufficient group almost double with 4.4%, and in the deficient group almost four times greater with a failure rate of 11.1%, although the chi-square test did not identify statistically significant differences (*p* = 0.105).

In the randomized clinical trial by Garg et al. [31], in which 32 dental implants were placed in the posterior mandible in 32 patients, patients were evenly divided into two groups: Group I with serum vitamin D levels <30 ng/mL measured on the day of diagnosis but supplemented for between 3 and 6 months with cholecalciferol 6000 IU/month; and Group II with serum vitamin D levels <30 ng/mL measured on the day of diagnosis. It was observed that in Group I the mesial crestal bone level decreased from 1.386 mm at the time of implant placement to 0.832 mm six months later and on the distal aspect from 1.310 mm to 1.085 mm, although this difference was not statistically significant. In Group II, the mesial crestal bone level decreased from 1.179 mm at the time of implant placement to 0.229 mm six months later and on the distal aspect from 1.065 mm to 0.285 mm, this difference being statistically significant (*p* < 0.01; 0.05). During the follow-up time, no dental implants were lost in either group.

In the randomized clinical trial by Kwiatek et al. [32] in which 122 patients received 122 dental implants placed in mandibular bicuspids and molars, the patients were divided into three groups: Group A with serum vitamin D levels <30 ng/mL measured on the day of dental implant placement, 6 weeks later, and 12 weeks later; Group B with serum vitamin D levels <30 ng/mL measured on the day of dental implant placement, 6 weeks later, and 12 weeks later, but supplemented during this time with vitamin D; and Group C with serum vitamin D levels ≥30 ng/m measured on the day of dental implant placement, 6 weeks later, and 12 weeks later. It was observed that, in Group A, mean bone level changed from 0.06 ± 0.48 at 6 weeks after implant placement to 0.08 ± 0.92 at 12 weeks although this change was not statistically significant (*p* < 0.232). In Group B, bone level went from 0.25 ± 0.51 at 6 weeks after implant placement to 0.53 ± 0.77 at 12 weeks, this difference being statistically significant (*p* < 0.001). In group C, the bone level went from 0.29 ± 0.53 at 6 weeks after implant placement to 0.48 ± 0.74 at 12 weeks with a statistically significant difference (*p* < 0.008). It was also observed that the bone level in Group B was significantly higher than in Group A (*p* < 0.028). During the follow-up period, no dental implants were lost in any of the groups.

The prospective three-cohort study by Tabrizi et al. [33] studied 90 patients who received 90 dental implants placed in mandibular molars. The patients were divided equally into three groups: Group 1 with serum vitamin D levels <10 ng/mL measured on the day of dental implant loading and 12 months later; Group 2 with serum vitamin D levels between 10–30 ng/mL measured on the day of dental implant loading and 12 months later; and Group 3 with serum vitamin D levels >30 ng/mL measured on the day of dental implant loading and 12 months later. Marginal bone loss was 1.38 ± 0.33 mm in Group 1; 0.89 ± 0.16 in Group 2; and 0.78 ± 0.12 in Group 3. Statistical analysis found significant differences in marginal bone loss between the three groups (*p* < 0.001) and a correlation was also observed between marginal bone loss and serum vitamin D levels (*p* < 0.001). During the follow-up period, no dental implants were lost in any of the groups.

## 4. Discussion

In its active form (1,25-dihydroxy vitamin D3), vitamin D plays a key role in the regulation of bone mineralization through the activation of osteoblasts and osteoclasts [34]. In addition, vitamin D has shown to have potent anti-inflammatory effects. As a result, it has been considered as an adjuvant therapy for many chronic diseases such as asthma, arthritis, and prostate cancer or psoriasis, among others [35].

Studies in North America and Europe have observed that vitamin D deficiency affects as much as 50% of the general population, so a large proportion of patients attending the dentist suffer from this hypovitaminosis [36,37]. When this type of deficiency exists, the medical consensus recommends supplementation with 400–800 IU/day of vitamin D to benefit bone and 400–2000 IU/day to achieve pleiotropic benefits, depending on age, weight, disease status, and ethnicity [38].

This systematic review aimed to evaluate dental implant osseointegration, marginal bone loss and survival rate as a function of serum vitamin D levels. A total of four studies met the inclusion criteria and included 1129 patients who received a total of 1984 dental implants, comparing bone remodeling during osseointegration, marginal bone loss, and/or dental implant survival between patients with normal and low serum vitamin D levels. The ratio between men and women observed in the data obtained is 1.07, so it does not seem that sex has an influence on the results obtained. With regard to age, the majority of studies include patients over 40 years of age, since it is at this age that vitamin D deficiencies begin to appear [29,31,32,33].

Regarding bone remodeling during osseointegration, two of the studies addressed this issue. It was observed that patients with vitamin D deficiency (<30 ng/mL) supplemented with vitamin D during the osseointegration phase underwent less bone reduction than patients who were not supplemented [31,32]. Furthermore, supplemented patients showed even less bone reduction than patients with adequate serum vitamin D levels (>30 ng/mL) [32]. These data are in agreement with studies conducted in animals, in which new bone formation around implants was found to improve with vitamin D supplementation [39,40,41]. At the same time, patients with adequate (>30 ng/mL) and insufficient (10–30 ng/mL) vitamin D levels are known to benefit less from vitamin D supplementation than patients with deficient serum levels (<10 ng/mL) [42].

Marginal bone loss was only measured in one of the studies reviewed, in which patients presenting sufficient serum vitamin D levels were found to undergo significantly less bone loss measured at one year compared with deficient serum vitamin D levels [33].

Comparing these data with a prior systematic review on marginal bone loss around implants [43], marginal bone loss in groups of patients with both sufficient and insufficient serum vitamin D levels remained within the expected physiological marginal bone loss, while marginal bone loss patients with deficient serum vitamin D levels surpassed physiological loss. Thus, it would appear that a vitamin D deficiency increases physiological marginal bone loss.

Regarding dental implant survival, in three of the articles reviewed, no implant losses occurred, although it should be noted that the follow-up periods in these studies were very short (6–12 months) [31,32,33]. Thus, it could be stated that serum vitamin D levels do not decrease early survival of dental implants. On the other hand, the study by Mangano et al. [29] with a follow-up reaching 14 years reveals more realistic survival percentages, whereby the survival of dental implants placed in patients with sufficient serum levels of vitamin D obtained 8% greater survival than those placed in patients with deficient vitamin D serum levels, although this difference was not statistically significant. The survival rate reported for the group with sufficient vitamin D levels was higher than that found in the most recent systematic review on long-term dental implant survival [44] (97.1% vs. 96.4%), but this was not true of of patients presenting insufficiency (95.6% vs. 96.4) or deficiency (88.9% vs. 96.4%). Moreover, it has been observed that dental implant failures decrease threefold when dental implants are placed by digitally guided surgery compared with freehand placement (2.25% vs. 6.42%) [45].

The present systematic review had some limitations, particularly the low number of studies included, and the lack of randomized clinical trials comparing normal and low vitamin D levels, in order to compare the implant results, having to include in this systematic review cohort studies that present less scientific evidence than randomized clinical trials. In addition, one of the RCTs had a high risk of bias due to the lack of data allocation, concealment, and randomization [31].

Thus, it would appear that presenting sufficient serum levels of vitamin D increases implant survival, although a greater number of randomized clinical trials with follow-up periods of several years are needed to confirm this finding.

## 5. Conclusions

Despite the limitations of this systematic review, it appears that serum vitamin D levels in patients may play a relevant role in osseointegration, marginal bone loss, and dental implant survival. For this reason, it may be advisable to determine the serum vitamin D level of each patient before placing dental implants and to provide vitamin D supplementation, when necessary, although this statement should be taken with caution as more randomized human clinical trials are needed to support this hypothesis.

## Figures and Tables

**Figure 1 ijerph-19-10120-f001:**
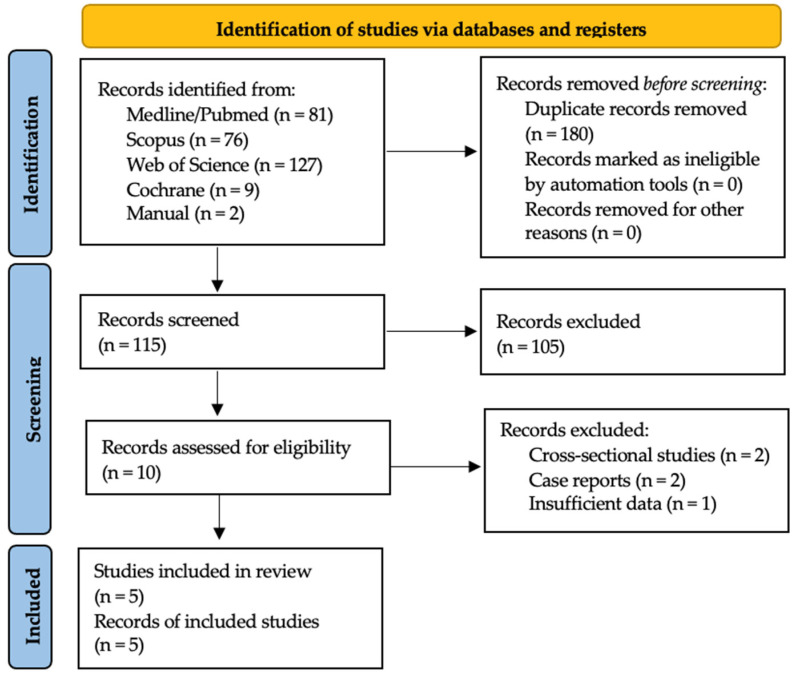
Flow chart illustrates the selection process.

**Table 1 ijerph-19-10120-t001:** Articles excluded and reason for exclusion.

Study	Reason for Exclusion
Wagner et al., 2017, Thim et al., 2022 [25,27]	Cross-sectional studies
Bryce et al., 2014, Fretwurst et al., 2016 [23,24]	Case report
Munhoz et al., 2019 [26]	Insuficient data

**Table 2 ijerph-19-10120-t002:** Information about selected studies including study type, number of patients, gender, mean age, vitamin D serum level, number and location of implants, implant survivals, marginal bone loss, and bone remodeling during osseointegration and follow-up.

Author and Year	Study	Patients (Number)	Gender (Male/Female)		Mean Age (Years)	Vitamin D Serum Level (Patients)	Timing of Vitamin D Sampling	Implants (Number)	Implants Location	Implants Survival (%)	Marginal Bone Loss (mm)/DM	Bone Remodeling during Osseointegration (mm)/DM	Mean Follow-Up (Months)
Mangano et al., 2018 [28]	Retrospective study. Three cohorts	885	455	430	57.3 ± 14.4	G1: <10 ng/mL: 27 G2: 10–30 ng/mL: 448 G3: >30 ng/mL: 410	Two weeks prior to surgery	1740	-	G1: 88.9 G2: 95.6 G3: 97.1	-	-	168
Garg et al., 2020 [30]	RCT	32	-	-	20–40 (range)	G1: <30 ng/mL (supplement): 16 G2: <30 ng/mL: 16	At the time of diagnosis. The subsequent blood samples were taken at 3-month and 6-month follow-up period from G1 patients	32	Mandibular posterior teeth	100	-	G1: M: 0.832 D: 1.085 G2: M: 0.229 D: 0.285 RVG	6
Kwiatek et al., 2021 [31]	RCT	122	57	65	43.8 ± 12.15	G1: <30 ng/mL (supplement): 48 G2: <30 ng/mL: 43 G3: >30 ng/mL: 31	On the day of surgery, after six weeks, and after twelve weeks.	122	Premolar and molar mandible	100	-	G1: 0.08 ± 0.93 G2: 0.53 ± 0.77 G3: 0.48 ± 0.74 RVG	3
Tabrizi et al., 2021 [32]	Prospective study. Three cohorts	90	56	34	G1: 41.50 ± 10.13 G2: 45.03 ± 11.16 G3: 40.73 ± 9.95	G1: <10 ng/mL: 30 G2: 10–30 ng/mL: 30 G3: >30 ng/mL: 30	At the time of loading and 12 months later	90	Molar mandible	100	G1: 1.38 ± 0.33 G2: 0.89 ± 0.16 G3: 0.78 ± 0.12/RVG	-	12

RCT: Randomized clinical trial; G: group; DM: Diagnostic method; M: mesial; D: Distal; RVG: radiovisiography.

**Table 3 ijerph-19-10120-t003:** Quality assessment of cohort studies using the Newcastle-Ottawa Scale.

	Selection				Comparability		Outcome			Number of Stars (Out of 9)
Study	S1	S2	S3	S4	C1	C2	E1	E2	E3	
Mangano et al., 2018 [28]	★	★	★	0	★	★	0	★	★	7
Tabrizi et al., 2021 [32]	★	★	★	0	★	★	0	0	★	6

★ = 1.

**Table 4 ijerph-19-10120-t004:** Quality assessment of included studies using the Cochrane bias assessment tool.

Study	Random Sequence Generation	Allocation Concealment	Blinding of Participants and Personnel	Blinding of Outcome Assessment	Incomplete Outcome Data	Selective Reporting	Other Bias
Garg et al., 2020 [30]	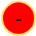	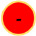	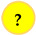	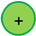	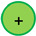	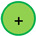	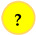
Kwiatek et al., 2021 [31]	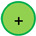	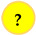	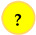	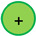	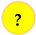	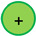	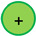

## Data Availability

Not applicable.

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
