# Peer review of "Influence of Serum Vitamin D Levels on Survival Rate and Marginal Bone Loss in Dental Implants: A Systematic Review"

_ijerph, 2022, doi:10.3390/ijerph191610120_

Round 1
Reviewer 1 Report
The manuscript is precisely presented and the design of the study adequate.
On this reviewer's concern Authors should improve the paragraph concerning the limits of the study, adding more information and critically pointing out the weakness of the work.
In addition the role of Vitamin D3 in the inflammation process should be developed in depth and further discussed.
According to this reviewer's consideration, publication of the present manuscript is suggested pending minor revision.
Author Response
-The manuscript is precisely presented and the design of the study adequate.
Response: Dear reviewer, thank you very much for the positive judgement of our manuscript. We proceed to explain your remarks.
- On this reviewer's concern Authors should improve the paragraph concerning the limits of the study, adding more information and critically pointing out the weakness of the work.
Response: The question is answered in lines 332-337.
- In addition the role of Vitamin D3 in the inflammation process should be developed in depth and further discussed.
Response: The anti-inflammatory capacity of vitamin D3 has been added in the text on lines 278-282.
-According to this reviewer's consideration, publication of the present manuscript is suggested pending minor revision.
Response: Thank you for your consideration
Reviewer 2 Report
A review of high quality both methodologically and in execution. The research question is clinically relevant. However, the data basis, only 4 original papers, is limited. In the preparation of the data, there are not more patient-specific parameters. In the discussion, these points should therefore be named more strongly (age, gender, etc.).
In the meantime, comparable reviews have been published. This should be changed accordingly in line 63.
In my opinion, the statement that the vitamin D level should be determined before implantation cannot be inferred from the results of the study. Also, the influence of vitamin D supplements is not really proven. Such statements should be made with great caution.
Author Response
-A review of high quality both methodologically and in execution. The research question is clinically relevant. However, the data basis, only 4 original papers, is limited. In the preparation of the data, there are not more patient-specific parameters. In the discussion, these points should therefore be named more strongly (age, gender, etc.).
Response: Thank you for your considerations. We have amended the text to expand on your comments on line 289-297.
-In the meantime, comparable reviews have been published. This should be changed accordingly in line 63.
Response: It has been modified as per your comment on lines 63-64.
-In my opinion, the statement that the vitamin D level should be determined before implantation cannot be inferred from the results of the study. Also, the influence of vitamin D supplements is not really proven. Such statements should be made with great caution.
Response: According to their considerations, the conclusions on lines 342-347 have been modified.
Reviewer 3 Report
The current study is very interesting an actual, presenting a systematic review regarding the influence of vitamin D levels on survival rate and marginal bone loss in dental implants. However, I would like to highlight several deficiencies:
- Introduction: inappropriate citations (line 55-60)- your state “numerous studies”- in citation there is a single one which does not refer to the subject.
- Material and methods: correct
- Results: please provide more detailed information on NOS scale (S1, … S4, C1, … E3)
- Discussion: - line 282- inappropriate citation, please replace.
§ Line 298- inappropriate citation, please replace.
§ Line 303- inappropriate citation, please replace.
§ Citation no. 44 is missing
- Conclusion: correct
Author Response
-Introduction: inappropriate citations (line 55-60)- your state “numerous studies”- in citation there is a single one which does not refer to the subject.
Response: citations have been modified from paragraph comprising lines 55-61.
-Material and methods: correct
Response: Thank you.
-Results: please provide more detailed information on NOS scale (S1, … S4, C1, … E3)
Response: The text on lines 193-200 has been modified to explain the NOS scale in more detail.
-Discussion: - line 282- inappropriate citation, please replace.
- Line 298- inappropriate citation, please replace.
- Line 303- inappropriate citation, please replace.
- Citation no. 44 is missing
Response: Citations have been modified in the text and in the bibliography. Thank you.
-Conclusion: correct
Response: Thank you.